# Simultaneous and independent topological control of identical microparticles in non-periodic energy landscapes

Nex C. X. Stuhlmüller [1], Farzaneh Farrokhzad [2], Piotr Kuświk [3], Feliks Stobiecki[3], Maciej Urbaniak [3], Sapida Akhundzada[4], Arno Ehresmann [4], Thomas M. Fischer [2] & Daniel de las Heras [1]

Topological protection ensures stability of information and particle transport against perturbations. We explore experimentally and computationally the topologically protected transport of magnetic colloids above spatially inhomogeneous magnetic patterns, revealing that transport complexity can be encoded in both the driving loop and the pattern. Complex patterns support intricate transport modes when the microparticles are subjected to simple time-periodic loops of a uniform magnetic field. We design a pattern featuring a topological defect that functions as an attractor or a repeller of microparticles, as well as a pattern that directs microparticles along a prescribed complex trajectory. Using simple patterns and complex loops, we simultaneously and independently control the motion of several identical microparticles differing only in their positions above the pattern. Combining complex patterns and complex loops we transport microparticles from unknown locations to predefined positions and then force them to follow arbitrarily complex trajectories concurrently. Our findings pave the way for new avenues in transport control and dynamic self-assembly in colloidal science.

The transport of microscopic colloidal particles suspended in fluids is relevant for a wide range of physical and biological phenomena including sedimentation[1], drug delivery[2–4], self-assembly[5–7], microfluidic devices[8–13], and active systems[14–16]. External fields are often used to control the motion of colloidal particles[17–19]. These include spatially uniform fields such as gravitational[20], electric[21], and magnetic[22–24] fields, as well as spatially inhomogeneous fields such as the manipulation of colloidal particles with optical tweezers[25]. Directed colloidal transport can be achieved via Brownian motors[26–28] that combine non-equilibrium fluctuations with spatially inhomogeneous energy landscapes[29–31].

Usually, the colloidal particles are transported along the same direction but the simultaneous transport of different particles across different directions is useful and even a requisite in systems of various length scales. For example, the transport of cargo on traffic networks requires organizing various subtasks simultaneously[32]. Sorting of microparticles driven on periodic lattices is possible because the particles travel along different directions depending on, e.g. their size[33–36]. In biology, the metabolism and structural diversity of the cell demand the regulation of a vast array of molecular traffic across intracellular and extracellular membranes.

In previous work, we have shown that robust, multidirectional, and simultaneous control of colloidal particles that differ in, e.g. their magnetic properties can be achieved with topological protection[37,38]. As illustrated in Fig. 1a, paramagnetic particles are placed above a

[1]Theoretische Physik II, Physikalisches Institut, Universität Bayreuth, D-95440 Bayreuth, Germany. [2]Experimatalphysik X, Physikalisches Institut, Universität Bayreuth, D-95440 Bayreuth, Germany. [3]Institute of Molecular Physics, Polish Academy of Sciences, 60-179 Poznań, Poland. [4]Institute of Physics and Center for Interdisciplinary Nanostructure Science and Technology (CINSaT), University of Kassel, D-34132 Kassel, Germany. ✉e-mail: delasheras.daniel@gmail.com

periodic magnetic pattern made of regions of positive and negative magnetizations normal to the pattern. A uniform external magnetic field of varying orientation drives the motion. The particles are transported following the minima of the periodic magnetic potential which results from the interplay between the complex but static field of the pattern and the simple but time-dependent uniform external field. The orientation of the magnetic field varies in time-performing loops. Hence, after one loop the orientation returns to its initial value. Loops that wind around specific orientations induce the transport of the colloidal particles by one unit cell of the magnetic pattern. During the loop, minima of the magnetic potential cross from one unit cell to the adjacent. Once the loop ends, the particle is in a position equivalent to the initial one but in a different unit cell. The motion is topologically protected in the sense that the precise shape of the loop is irrelevant. Only the set of winding numbers of the modulation loop around the specific orientations (the topological invariant) determines the transport direction. The motion is therefore robust against perturbations.

The specific orientations of the external field that are relevant to control the motion depend on both the symmetry of the pattern[37] (e.g. square vs. hexagonal) and the particle properties. Hence, particles with different properties, e.g. paramagnetic and diamagnetic particles above hexagonal magnetic patterns[39] as well as micro-rods of different lengths[38], can be transported in different directions independently and simultaneously using periodic patterns. However, the use of periodic patterns imposes several limitations on the transport. All particles that belong to the same topological class (e.g. identical paramagnetic particles or rods of the same length) are transported along the same direction, independently of their absolute position above the pattern as schematically represented in Fig. 1a. In addition, the location of the particles above the pattern is unknown a priori and it must be determined externally via, e.g. direct visualization via microscopy.

These limitations are overcome here using inhomogeneous (non-periodic) patterns. We make either the symmetry, Fig. 1b, or the global orientation, Fig. 1c, of the magnetic pattern dependent on the absolute position above the pattern. As a result, the specific orientations of the

external field that control the motion depend also on the space coordinate. The direction of the transport can then be locally controlled by the modulation loop of the external field and also via the local symmetry of the inhomogeneous magnetic pattern. We can imprint the complexity of the transport mainly to the pattern, and then use simple loops to generate complex transport as illustrated in Fig. 1b. Following this idea we create non-periodic patterns that transport the particles to a desired position by just repeating simple modulation loops. We also create patterns in which the colloidal particles follow arbitrarily complex trajectories driven by a simple time-periodic modulation loop. Additionally, we create simple patterns and encode the complexity of the transport in the modulation loops as sketched in Fig. 1c. This allows us to simultaneously and independently control the transport of identical colloidal particles located at different positions above the pattern. We design for example a complex modulation loop that controls the transport of 18 identical colloidal particles individually and simultaneously. Beyond its fundamental interest, our work opens a new route to control the transport in colloidal systems with potential applications in reconfigurable self-assembly[40–43].

## Results

The plane in which the particles move (action space) splits into allowed and forbidden regions. In the allowed (forbidden) regions the stationary points of the magnetic potential are minima (saddle points). The boundaries between allowed and forbidden regions in action space are the fences. The position of the fences in control space $\mathcal{C}$ (a sphere that represents all possible orientations of the external field) depends on the symmetry of the pattern and it determines the loops that induce colloidal transport (see Fig. 1). An extended summary of the transport in periodic patterns[37] is provided in Supplementary Note 1 and Supplementary Figs. 1 and 2.

Here we focus on transport in inhomogeneous patterns. Sophisticated transport modes can be achieved by adding complexity to either the patterns, the loops, or to both of them. We see examples of each type in the following sections. Details about the experiments and computer simulations are given in the "Methods" section.

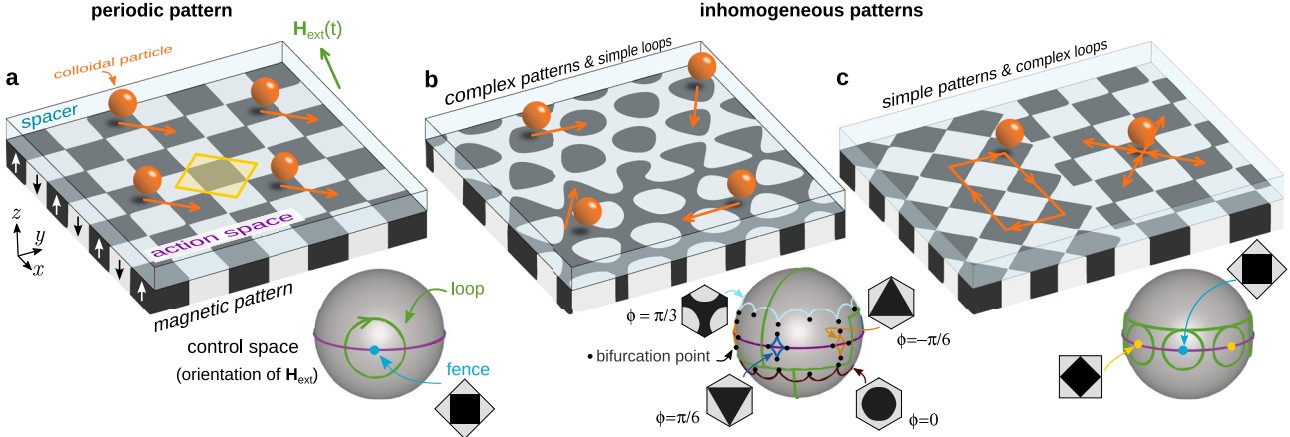

**Fig. 1 | Periodic vs inhomogeneous patterns. a** Periodic square pattern (a unit cell is highlighted in yellow), **b** hexagonal pattern in which the symmetry phase $\phi$ varies in space, and **c** a pattern made of two square patterns rotated by an angle of 45°. The patterns are made of regions with positive (black) and negative (white) magnetization normal to the pattern, see vertical arrows in (**a**). A polymer coating protects the patterns and acts as a spacer for the paramagnetic colloidal particles (orange) that are suspended in a solvent and move in a plane parallel to the pattern (action space). The motion is driven by a uniform external field (green arrow). The control space $\mathcal{C}$ (gray spheres) represents all possible orientations of the external field. The orientation of the external field varies in time performing a loop (green curves). Loops that wind around special orientations induce particle transport. These special orientations are determined by the position of the fences and

bifurcation points in control space which depend on the local symmetry of the pattern. Shown are the fences of square patterns for one (**a**) and two (**c**) different orientations, as well as those of four hexagonal patterns with different symmetry phases $\phi$ (**b**). We also indicate the bifurcation points (black circles) in (**b**) which are those points where two fence segments meet. Next to the fences, we show the corresponding unit cell of the pattern. In periodic patterns (**a**) all the particles move in the same direction (orange arrows), independently of their position above the pattern. In inhomogeneous patterns, a single modulation loop can induce transport in different directions depending on the position of the particle above the pattern. Complex particle trajectories can be generated using complex patterns and simple loops (**b**) or simple patterns and complex loops (**c**).

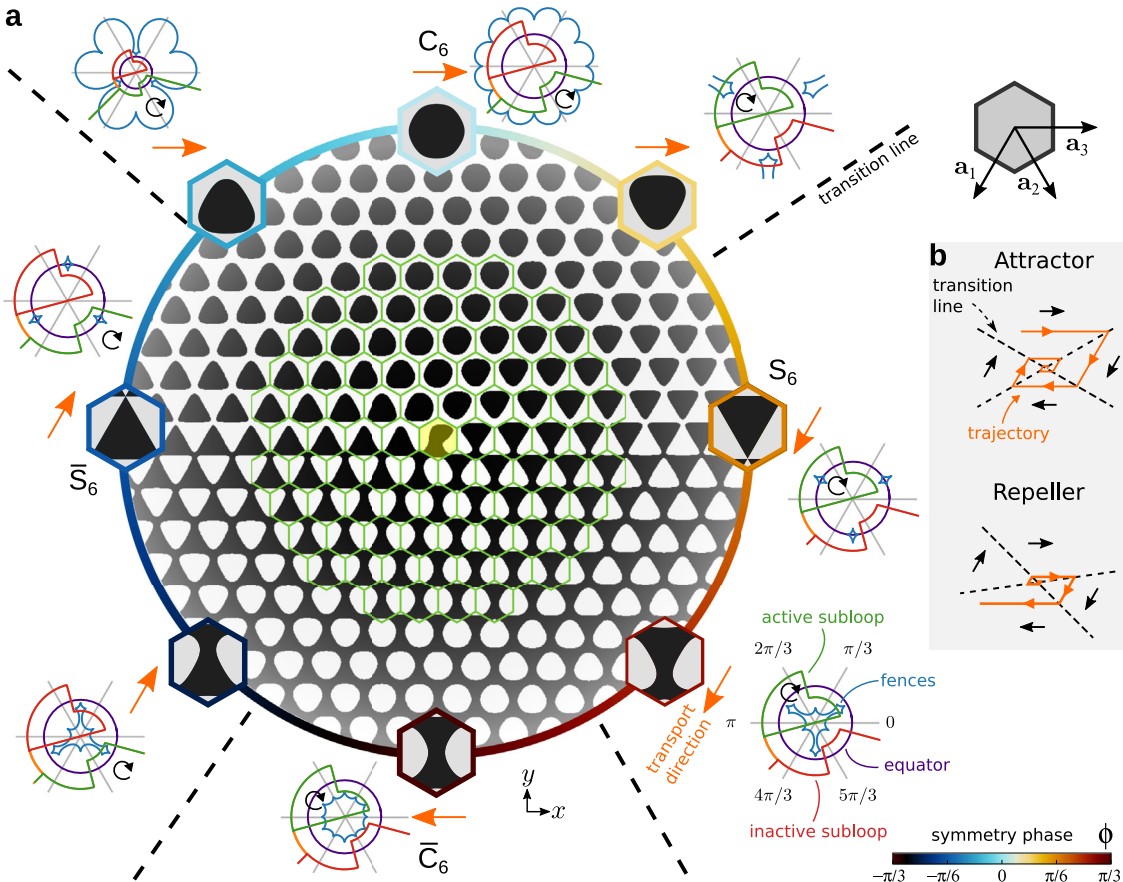

**Fig. 2 | Pattern with a topological defect. a** Magnetic pattern with a topological defect in the symmetry phase $\phi$. The pattern is dissected into hexagonal cells (green hexagons). The central cell (yellow) contains the defect. Enlarged Wigner–Seitz cells of selected periodic hexagonal lattices with symmetry phase $\phi$ (see color bar) corresponding to their position in the pattern are shown. Next to each enlarged cell, we plot a stereographic projection of the corresponding control space and the modulation loop that attracts the particles toward the defect. Shown are the fences (blue), the equator (violet), and both the active (green) and the inactive (red) subloops. The loop winds as indicated by the circular black arrow. The two apparently open segments of the loop are actually joined at the south pole of the control space (not visible due to the projection). The transport direction (orange arrows) changes at the transition lines (black-dashed lines). **b** Illustrative configurations of the position of transition lines (black-dashed lines) that give rise to particle trajectories moving towards the defect (attractor) or away from it (repeller). The particle trajectories are illustrated in orange.

## Complex patterns and simple loops

There is a full family of periodic hexagonal patterns characterized by the value of the symmetry phase $\phi$ (see the "Methods" section) and illustrative examples in Fig. 1b. We render the symmetry phase a continuous function of the position, which creates an inhomogeneous symmetry phase modulated pattern such as the example in Fig. 1b. For slow enough spatial changes of the symmetry phase, the cells of the modulated pattern deviate only weakly from the Wigner–Seitz cells of corresponding periodic patterns with fixed values of the symmetry phase. Hence, knowing how to control the transport in periodic patterns is enough to control the transport in inhomogeneous situations.

We focus first on complex inhomogeneous patterns designed to achieve locally different transport for a single specific task. Most of the complexity of the transport is embedded in the pattern and therefore the modulation loops of the external field are simple.

## Topological defect in the symmetry phase

We show in Fig. 2 a symmetry phase modulated hexagonal pattern. The details to generate the pattern are given in the "Methods" section. Each time we wind around the center of the pattern we go through the full family of hexagonal patterns exactly once (including the inverse patterns with opposite magnetization) and return to the initial symmetry phase. This introduces a topological defect at the center of the pattern where the symmetry phase is not well defined.

The symmetry phase is constant along radial directions and the modulation is weak everywhere except near the defect. To illustrate this, we have dissected the pattern into hexagonal cells in Fig. 2a. We also show enlarged Wigner Seitz cells of periodic patterns with a symmetry phase corresponding to that of the radial ray of the inhomogeneous pattern. The Wigner Seitz cells of the periodic patterns resemble closely the cells of the inhomogeneous pattern, even in the proximity of the central defect. It is therefore expected that the transport in the inhomogeneous pattern can be understood in terms of the transport in periodic patterns.

The location of the fences in the control space varies substantially as we wind around the defect in the action space. (See the stereographic projections of control space for selected values of the symmetry phase in Fig. 2a and Supplementary Fig. 1.) Hence, it is possible to transport the microparticles into different directions depending on the sector of the pattern. In particular, we can construct modulation loops that use the central defect of the pattern as either an attractor or a repeller of colloidal particles.

A stereographic projection of the modulation loop that attracts the particles towards the defect is shown in Fig. 2a next to each enlarged Wigner–Seitz cell. The loop is made of two subloops. Only one of the subloops is active (green) for each value of the symmetry phase $\phi$. The subloop is active in the sense that it induces net transport for those particles located in sectors of the pattern with

that value of $\phi$. The other subloop is inactive (red) in the sense that after one complete subloop, the particle returns to its position and hence there is no net transport. Using two subloops we control simultaneously the transport direction in sectors of the pattern with opposite magnetization (different values of the symmetry phase). Note for example how the active subloop in regions with $C_6$ symmetry ($\phi = 0$) becomes the inactive subloop in those regions with an inverse pattern $\overline{C}_6$ ($\phi = \pm\pi/3$) and vice versa (see Fig. 2a). To induce transport a subloop must wind around at least three bifurcation points of the fences in control space $\mathcal{C}$, as explained in the Supplementary Note 1. Recall that control space is simply the surface of a sphere in which each point corresponds to one orientation of the external magnetic field. The bifurcation points are the points in which two segments of the fences meet in $\mathcal{C}$, see examples in Supplementary Fig. 2.

The complete attractor loop, made of two subloops, induces four different transport directions (along $\pm\mathbf{a}_1$ and along $\pm\mathbf{a}_3$) depending on the value of the symmetry phase (see Fig. 2a). Here, $\mathbf{a}_i$, $i = 1, 2, 3$ are three lattice vectors of the periodic hexagonal pattern (see Fig. 2 and the "Methods" section). The transition between the different transport directions, e.g. from $+\mathbf{a}_3$ to $-\mathbf{a}_1$, occurs at specific values of the symmetry phase that can be adjusted with the loop. See the transition lines (dashed-black lines) in Fig. 2a.

By controlling the location of the transition lines we fix whether the defect acts as an attractor or a repeller of particles (see Fig. 2b). In both cases, the particles wind clockwise around the defect. Instead of changing the position of the transition lines, we could also control whether the defect attracts or repels microparticles by reversing the direction of the transport. However, this requires a complete redesign of the modulation loop. Simply reversing the direction of the modulation loop does not reverse in general the transport direction in the whole pattern due to the occurrence of non-time reversal ratchets in hexagonal patterns[37,39].

In Fig. 3a, b we show the trajectories of a colloidal particle located above the defect pattern according to Brownian dynamics simulations. The particle is randomly initialized above the pattern and then subjected to several repetitions of the attractor loop shown in Fig. 2. We also show the trajectory followed by the particle under the repetition of the repeller loop, depicted in Fig. 3c. The repeller and the attractor loops have similar shapes since they differ only in the values of $\phi$ at which the transport direction changes. The corresponding experimental trajectories are shown in Fig. 3d. In the experiments, there are several colloidal particles that are initially located above the pattern in random positions. If the attractor loop is repeated enough times, one colloidal particle will have reached the defect with almost certainty. Once a particle reaches the defect it stays there. In the experiments, further colloidal particles that try to enter the defect are repelled by the particle already occupying the center via dipolar repulsion. We can thus use the attractor loop to initialize one microparticle in the defect center. Whereas the location prior to the action of the attractor loop was unknown, it is known after the repeated application of the loop. The topological initialization is robust to thermal fluctuations. Brownian dynamics simulations of colloidal particles at higher temperatures still initialize the location of the defect. We briefly discuss the effect of finite temperature in the "Methods" section and Supplementary Fig. 4.

### Encoding complex trajectories in the pattern

Patterns with spatial modulation of the symmetry phase can be used to encode arbitrarily complex particle trajectories. The patterns are designed to induce the desired trajectory when the particles are subjected to the repetition of a simple modulation loop of the orientation of the external field. The modulation loop transports particles along all possible directions in hexagonal patterns, i.e. along $\pm\mathbf{a}_i$ with $i = 1, 2, 3$, but in a way that only one direction is active for a given value of the symmetry phase. For example, particles on top of regions with $C_6$ symmetry are transported towards $-\mathbf{a}_3$. The transport direction

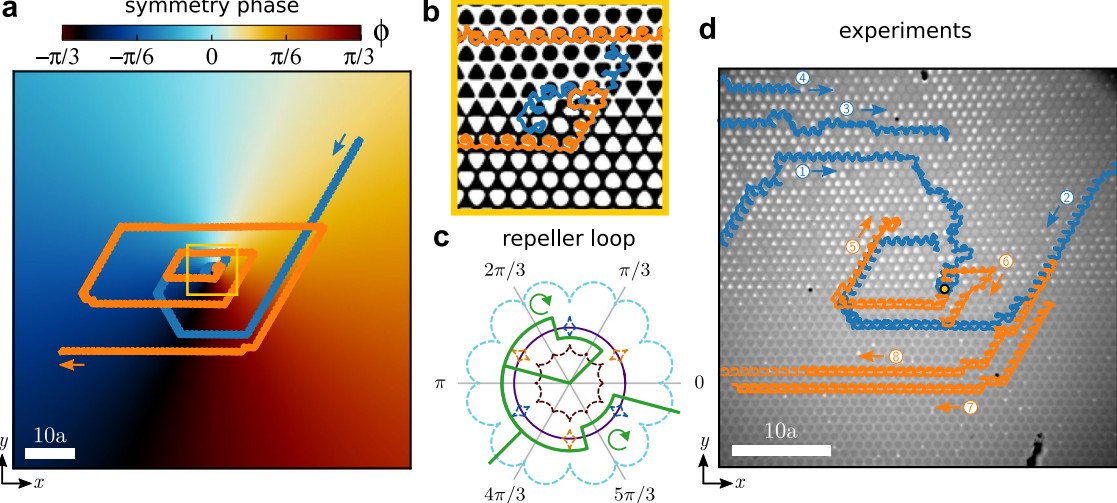

**Fig. 3 | Attractor and repeller of particles. a** Trajectory of a colloidal particle (randomly initialized) obtained with Brownian dynamics simulations above a pattern with a central topological defect in the symmetry phase. The blue (orange) trajectory is generated by the repetition of the attractor (repeller) modulation loop that moves particles towards (away from) the defect. The pattern is colored according to the value of the symmetry phase (color bar). The scale bar is $10a$. **b** Close-up of the region indicated by a yellow square in (**a**) and the trajectories around the central defect. The background shows the local magnetization of the pattern. **c** Stereographic projection of the repeller loop (green) in $\mathcal{C}$. The equator (violet circle) and the fences of the $C_6$ and $S_6$ patterns as well as their inverse patterns, $\overline{C}_6$ and $\overline{S}_6$, (dashed curves) are also depicted as a reference. The fences are colored according to the value of the symmetry phase. The two apparently open segments of the loop are actually joined at the south pole of the control space (not visible due to the projection). The loop is made of two subloops winding clockwise, as indicated by the circular arrows. **d** Experimental trajectories of several colloidal particles (labeled with a numbered circle) above the same pattern with a topological defect (yellow circle). The trajectories induced by the attractor (repeller) loop are colored in blue (orange). Blue and orange trajectories correspond to different experiments and have been superimposed in the figure. Note that under the microscope regions with negative magnetization appear darker than regions with positive magnetization, i.e. the opposite of our color choice in e.g. (**b**). The scale bar is $10a$ and the lattice constant of one cell is approx. 14 μm. Movies of the simulated and the experimental motion are provided in Supplementary Movie 1.

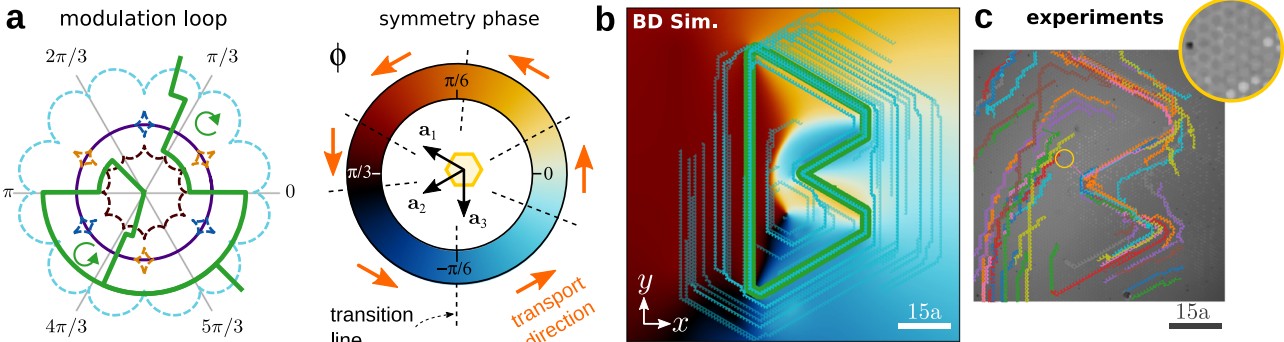

**Fig. 4 | Symmetry phase modulated pattern. a** Stereographic projection of control space showing the equator (violet circle), the closed modulation loop (green-solid curve), and the fences of patterns with $C_6$, $S_6$, $\overline{C_6}$ and $\overline{S_6}$ symmetries (dashed curves). The two apparently open segments of the loop are actually joined at the south pole of the control space (not visible due to the projection). The fences are colored according to the value of the symmetry phase (see the annular color bar). The transport directions induced by the loop (orange arrows) change at specific values of the symmetry phase $\phi$ as indicated by the transition lines (black-dashed lines). **b** Symmetry phase modulated pattern (the color indicates the value of the symmetry phase). A global rotation, $\psi = \pi/2$ in Eq. (6), makes one transport direction (lattice vector $\mathbf{a}_3$) parallel to the vertical axis. Particles above the pattern and subjected to the repetition of the modulation loop in (**a**) write the letter "B".

Thin cyan lines show simulated particle trajectories for randomly initialized particles above the pattern. After several repetitions of the modulation loop, most particles enter the stable trajectory, highlighted with a thick green-solid line. **c** Experimental trajectories of colloidal particles above the pattern depicted in (**b**) and subjected to the modulation loop shown in (**a**). The region shown in the experiments (**c**) is smaller than that in simulations (**b**) due to the field of view of the microscope. The inset in (**c**) is a close view of the region indicated with a yellow circle in which we have altered the contrast of the image to better visualize the magnetization. Under the microscope regions with negative magnetization appear darker than those with positive magnetization. A colloidal particle (black dot) is also visible in the inset. A movie of the motion in simulations and experiments is provided in Supplementary Movie 2.

changes at specific values of the symmetry phase determined by the modulation loop. In Fig. 4a we show the modulation loop together with the representative fences and the resulting transport direction for each value of the symmetry phase.

The detailed procedure to generate the patterns is described in the "Methods" section and Supplementary Fig. 3. In essence, we first draw the trajectories that the particles should follow by hand. Then, at each position along the trajectory, we encode the transport direction using the value of the symmetry phase. Finally, the value of the symmetry phase at each point in the complete pattern is calculated as a spatially resolved weighted average of the symmetry phase along the trajectory. As a result, the symmetry phase varies smoothly across the pattern except for the occurrence of string-like topological defects in the symmetry phase.

Figure 4b shows a symmetry phase modulated pattern together with the corresponding simulated particle trajectories. The value of the symmetry phase is color-coded (see color bar). The pattern is designed to transport the particles along one stable trajectory that forms a closed loop resembling the letter "B". In Fig. 4b we have highlighted the stable trajectory with a thick green line. Most particles above the pattern either enter the stable trajectory or leave the pattern. Occasionally one particle gets stuck in specific regions of the pattern. This can potentially be avoided by the introduction of random fluctuations in the modulation loop. In the presence of strong Brownian motion, the stable trajectories broaden to a width of a few unit cells, and additional stable trajectories might occur.

Corresponding experimental trajectories are shown in Fig. 4c. Even though the agreement is not perfect, the experimental trajectories follow closely the prescribed letter "B", demonstrating, therefore, the potential of the method. Small variations in the position of the fences due to the imperfections of the pattern are likely the reason behind the deviations shown in the experiments. Fine-tuning the modulation loop and the height of the particles above the pattern would likely improve the results.

## Simple patterns and complex loops

We follow now the opposite approach by encoding the complexity in the modulation loop. We create simple inhomogeneous patterns by concatenating large patches of periodic square patterns. The patches differ in the global orientation of the lattice vectors given by a global phase $\psi$ (see the "Methods" section). Each (simple) patch allows for a rich variety of transport tasks. The task in each patch can be controlled individually and simultaneously using rather complex modulation loops in control space.

The fences of the $C_4$ square pattern are four equidistant points on the equator (see Supplementary Note 1). The four fence points in $\mathcal{C}$ correspond to external fields pointing along the positive and negative directions of the lattice vectors[37,44], i.e. along $\pm\mathbf{a}_1$ and $\pm\mathbf{a}_2$. Therefore, rotating the lattice vectors also rotates the position of the fences in control space. Thus, it is possible to construct loops that wind around different fences in $\mathcal{C}$, and hence induce different transport directions, depending on the orientation of the pattern $\psi$. An illustration is shown in Fig. 5a.

Since the fences are points in $\mathcal{C}$ it is in principle possible to concatenate an arbitrarily large number of patches with different orientations and control the motion in each of them independently. In practice, limiting factors might appear due to e.g. imperfections in the patterns that effectively make the fences in $\mathcal{C}$ extended regions, the angular resolution with which the orientation of the external field can be controlled, and the presence of Brownian motion. Due to the limiting factors, two patterns can be resolved independently if they are rotated by an angle of at least $\Delta\psi$. Hence, the maximum number of patches that can be controlled independently is $(\pi/2)/\Delta\psi$ since after a rotation of $\pi/2$ a $C_4$ pattern repeats itself (and so do the fences).

With a resolution $\Delta\psi = 5°$ it is then possible to control the motion in up to 18 patches independently. As an example we program a single loop in $\mathcal{C}$ that writes the first eighteen letters of the alphabet simultaneously, (see Fig. 5b and Supplementary Movie 3). Note that the letters are rotated by an angle $\psi$. For simplicity, we have designed an algorithm to write custom trajectories in a square pattern with global orientation $\psi = 0$. Next, we apply a global rotation to the modulation loop to control the transport in patterns with a generic orientation $\psi$. As a result, the trajectories are also rotated.

The loop that writes the first 18 letters of the alphabet contains 2086 simple commands. Each command is a small closed subloop that

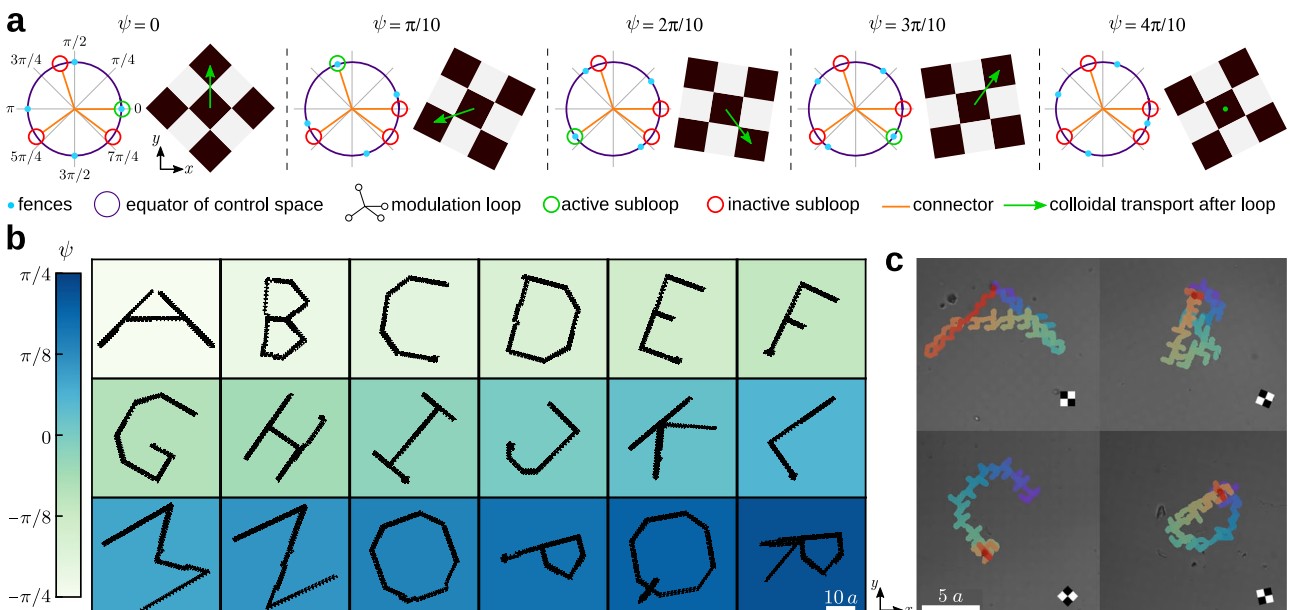

**Fig. 5 | Simple patterns and complex loops. a** Five square magnetic patterns (and their corresponding control spaces) with a different value of the global orientation $\psi$, as indicated. The fences in $\mathcal{C}$ (blue circles) are four points located on the equator (violet circle). The position of the fences depends on the value of $\psi$. The modulation loop consists of four interconnected subloops that wind counterclockwise. A subloop is active (green) if it winds around a fence point (blue circles) and inactive (red) otherwise. The orange segments of the modulation loop simply connect the different subloops. Depending on the value of $\psi$, the modulation loop induces different transport directions (green arrows) or no transport at all. **b** A pattern made of 18 patches with square symmetry and different global orientation $\psi$ (color bar). A modulation loop controls the trajectories of particles above each patch simultaneously and independently. The particle trajectories (black) write the first 18 letters of the alphabet. The length of the scale bar is 10$a$. A movie can be found in Supplementary Movie 3. **c** Experimental trajectories of colloidal particles above four square patches rotated with respect to each other. A schematic unit cell illustrating the global orientation is depicted in each patch. The length of the scale bar is 5$a$ and in this case, we use patterns with $a = 7$ μm. A unique modulation loop transports the four colloidal particles simultaneously. The trajectories are colored according to the time evolution from blue (initial time) to red (final time). A movie showing the whole time evolution and a one-to-one comparison with computer simulations is available in Supplementary Movie 4.

either transports a particle in one unit cell along the four possible directions of the square lattice or leaves the particle in the same position, similar to the loops in Fig. 5a. Even though an angular resolution of $\Delta\psi = 5°$ is achievable experimentally, the number of commands required by the complete loop exceeds our current experimental capabilities. Nevertheless, we show in Fig. 5c, the experimental trajectories of a simplified loop that writes low-resolution versions of the first four letters of the alphabet. The loop is made of 96 simple commands. The agreement with computer simulations is essentially perfect, as we demonstrate in a one-to-one comparison in Supplementary Movie 4.

The simultaneous control of the transport in several patches of rotated square patterns is particularly simple due to the simplicity of the fences in $\mathcal{C}$. However, the same ideas apply to patterns with other symmetry classes.

Here, we have initialized the particles in the desired positions within their respective patches. As we discuss now, it is possible to automate this process by combining the patches with complex patterns.

**Complex patterns and complex loops**
Complete control over the colloidal transport is achieved by combining complex patterns and complex loops. In Fig. 6 we combine three C$_4$ patches that differ in their global orientation $\psi$ and three hexagonal patterns with a topological defect in the symmetry phase. The transition between both patterns occurs smoothly within a region of length equivalent to approximately five unit cells of the square patterns.

We first make use of the patterns with a topological defect to move randomly placed particles toward the defects. We simply repeat the attractor modulation loop shown in Fig. 2 several times such that the particles move and stay at the defects, see the blue trajectories of

the particles in Fig. 6. Once this initialization stage is finished we know the precise position of the particles and can control them independently. Using two simple loops we transport the particles downwards from the defects to the square patches. We use one loop to move the particles in the defect pattern (orange trajectories) and another loop to move the particles in the transition region and the square patches (green trajectories). Then, a relatively complex loop controls the motion of the three particles independently. Each particle follows a complex trajectory drawing either a square, a triangle, or a cross depending on the value of the global orientation $\psi$ (red trajectories). Experimentally we tested each part of the loop separately, as shown in the insets of Fig. 6. Again, the agreement between simulations and experiments is excellent. The small errors that occur in the experimental trajectories, likely due to imperfections in the pattern, do not affect the global shape of the trajectories. A movie of the whole process is shown in Supplementary Movie 5.

## Discussion
We have shown that the combination of a complex static magnetic field with a simple time-dependent uniform external field of varying orientation allows us to control the motion of several identical microparticles independently and simultaneously. The transport complexity can be broken down to a finite set of special orientations of the external field. A modulation loop that winds around one of those orientations induces transport along a known direction in a known region of the pattern. The motion is topologically protected since only the winding numbers of the modulation loop around the special orientations (topological invariant) are important. Hence, it is relatively simple to generate loops and patterns that induce arbitrarily complex trajectories. Our ideas might be transferable to other systems

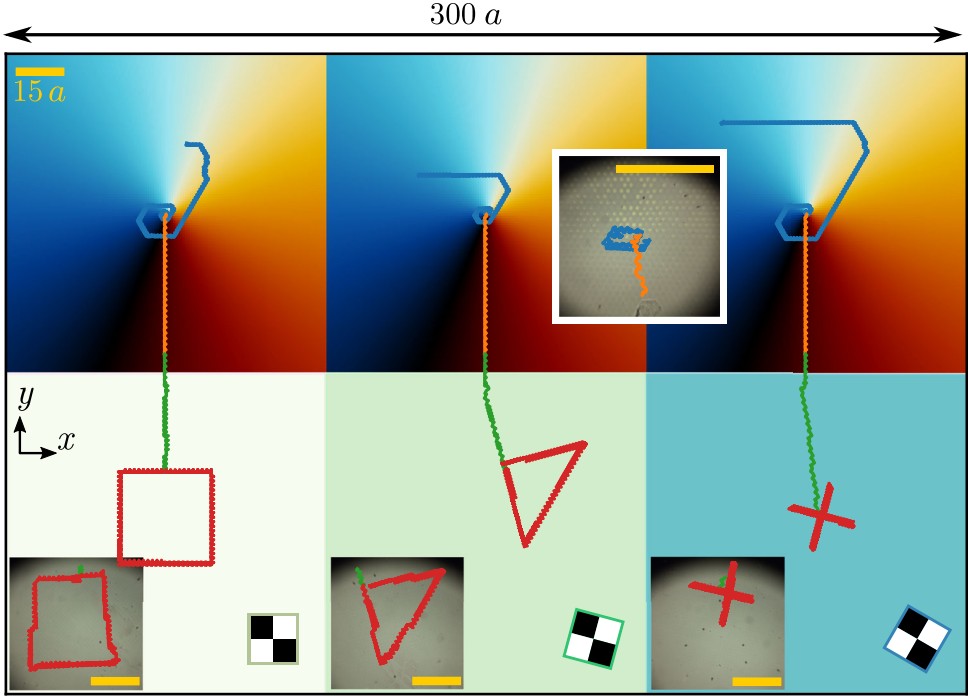

**Fig. 6 | Complex patterns and complex loops.** Brownian dynamics simulations of the transport of colloidal particles above a complex pattern made of three patches, each one with a topological defect in the symmetry phase (top) connected to three patches with square symmetry (down) rotated with respect to each other. The color of the patches with topological defects indicates the value of the symmetry phase $\phi$. The color of the square patches indicates the global rotation $\psi$, illustrated with a sketch of the magnetization. A unique complex modulation loop made of four parts drives the transport in the whole system. In the first part, the repetition of the attractor loop moves the particles toward the defects (blue trajectories) and lets them wait there. The second part of the loop moves the particles downwards through the patterns with defects (orange trajectories). The third part of the loop moves the particles downwards in the square patterns (green trajectories). The last part of the loop writes a custom trajectory (square, triangle, and cross) depending on the global orientation $\psi$ of the pattern (red trajectories). Insets show the corresponding experimental trajectories. The length of the scale bars (yellow) is 15$a$.

in which the transport is also based on topological protection. These include, solitons[45], nano-machines[46,47], sound waves[48,49], photons[50,51], and quantum mechanical excitations[52].

The complexity of the transport is encoded in the magnetic potential which varies in space and in time via the magnetic patterns and the modulation loops, respectively. An alternative approach that encodes the transport in the particle shape has appeared recently[53]. There, Sobolev et al. find the shape of the rigid body that traces the desired trajectory when rolling down a slope. We have restricted our study to identical isotropic paramagnetic particles. However, as discussed in the "Introduction" section, colloidal particles with different characteristics (e.g. diamagnetic and paramagnetic particles or particles with different shapes) might belong to different topological classes. The fences of particles belonging to different topological classes are located in different regions in $\mathcal{C}$. Above non-periodic patterns, the control space of particles belonging to different topological classes will also depend on the space coordinate. A precise control over the transport depending not only on the position but also on the particle characteristics is then possible. Therefore, beyond offering the possibility to control the transport of identical microparticles simultaneously, our work also opens a new route towards dynamical self-assembly in colloidal science. As an example, we have created a colloidal rod factory[54] in which identical isotropic particles are transported toward a reaction site in which they self-assemble. Only when they reach the desired aspect ratio, do the rods leave the polymerization site following the desired trajectory. The use of patchy colloids[55–58] with, e.g. hybridization of complementary DNA strands[59–61] and other shape-anisotropic particles[62,63] would offer more versatility to create complex functional structures.

We have considered transport above patterns made of identical patches rotated with respect to each other. It is also possible to combine patches of patterns with different symmetries provided that their respective fences do not overlap in control space. Moreover, a combination of both, i.e. a pattern made of patches with different symmetries, e.g. $C_4$ and $C_6$, that in addition are rotated with respect to each other would substantially increase the number of tasks that can be done simultaneously since their respective fences in control space do not overlap.

In the experiments, the Brownian motion of the colloidal particles is negligible but it might play a role in other systems with smaller colloids and/or at higher temperatures. Since the transport is topologically protected, it is robust against perturbations such as the presence of Brownian motion[44]. If we make Brownian motion more prominent (e.g. by increasing the temperature or reducing the particle size) the particles start to deviate from the expected trajectories but overall the transport is robust. An example of Brownian dynamics simulations at different temperatures is shown in Supplementary Fig. 4. The topological protection will disappear due to Brownian motion at sufficiently high temperatures and for small enough particles. A possible solution would then be to increase the magnitude of either the pattern field or the external magnetic field such that the magnetic forces dominate again the transport.

Our systems are very dilute and therefore direct interparticle interactions and hydrodynamic interactions do not play any role. However, it would be interesting to look at the effect of both super-adiabatic forces[64] and long-range hydrodynamic interactions[65] in denser systems.

## Methods

### System setup and computer simulations

Identical paramagnetic colloidal particles immersed in a solvent are located above a magnetic pattern and are restricted to move in a plane parallel to the pattern ($xy$-plane), which we call action space $\mathcal{A}$ (Fig. 1a). The pattern contains regions of positive, $+m_\mathrm{p}$, and negative, $-m_\mathrm{p}$, uniform magnetization along the $z$-direction (normal to the pattern). The width of the domain walls between oppositely magnetized regions is negligible. The particles are driven by a time- and space-dependent external magnetic potential $V_\mathrm{mag}(\mathbf{r}_\mathcal{A},t)$. The potential is generated by the static but space-dependent magnetic field of the pattern $\mathbf{H}_\mathrm{p}(\mathbf{r}_\mathcal{A})$ and a time-dependent but spatially homogeneous external magnetic field $\mathbf{H}_\mathrm{ext}(t)$. Here $\mathbf{r}_\mathcal{A}$ is the space coordinate in action space and $t$ denotes the time. The magnitude of the external field (constant) is much larger than that of the pattern field, i.e. $H_\mathrm{ext} \gg H_\mathrm{p}(\mathbf{r}_\mathcal{A})$ for any position in $\mathcal{A}$. Hence, the magnetic potential, which is proportional to the square of the total magnetic field $V_\mathrm{mag} \propto -(\mathbf{H}_\mathrm{ext} + \mathbf{H}_\mathrm{p}) \cdot (\mathbf{H}_\mathrm{ext} + \mathbf{H}_\mathrm{p})$, is dominated by the coupling between the external and the pattern fields:

$$V_\mathrm{mag}(\mathbf{r}_\mathcal{A},t) \approx -\upsilon_0 \chi \mu_0 \mathbf{H}_\mathrm{p}(\mathbf{r}_\mathcal{A}) \cdot \mathbf{H}_\mathrm{ext}(t). \tag{1}$$

Here $\mu_0$ is the vacuum permeability, $\chi$ is the magnetic susceptibility of the colloidal particle, and $\upsilon_0$ is the particle volume[37]. We have omitted the term proportional to $\mathbf{H}_\mathrm{ext} \cdot \mathbf{H}_\mathrm{ext}$ in $V_\mathrm{mag}$ since it is just a constant and therefore it does not affect the motion.

In the overdamped limit, the equation of motion of one particle reads

$$\gamma \dot{\mathbf{r}}_\mathcal{A} = -\nabla_\mathcal{A} V_\mathrm{mag} + \boldsymbol{\eta}, \tag{2}$$

where $\gamma$ is the friction coefficient against the implicit solvent, the overdot denotes time derivative, $\nabla_\mathcal{A}$ is the derivative with respect to $\mathbf{r}_\mathcal{A}$, and $\boldsymbol{\eta}$ is a delta-correlated Gaussian random force with zero mean that models the effect of the collisions between the molecules of the solvent and the colloidal particle (Brownian motion). We define our energy scale $\varepsilon$ as the absolute value of the average external energy that a particle would have when the external magnetic field points normal to the pattern. Hence, absolute temperature $T$ is given in reduced units $k_\mathrm{B}T/\varepsilon$ where $k_\mathrm{B}$ is the Boltzmann's constant. We use the magnitude of a lattice vector of the periodic pattern $a$ as the length scale. The timescale is hence given by $\tau = \gamma a^2/\varepsilon$. We use adaptive Brownian dynamics[66] to efficiently integrate the equation of motion. In the experiments, the magnetic forces strongly dominate over the random forces. Hence, random forces do not play any role. We use Brownian dynamics simulations due to the overdamped character of the motion in the viscous aqueous solvent. The code to simulate the colloidal motion and to generate the modulation loops is available via Zenodo[67].

As the external magnetic field is homogeneous in space, it can be solely described by its orientation. The set of all possible orientations of $\mathbf{H}_\mathrm{ext}$ forms a spherical surface that we call control space $\mathcal{C}$. A point in $\mathcal{C}$ indicates an orientation of $\mathbf{H}_\mathrm{ext}$. We drive the colloidal motion by performing closed loops of the orientation of $\mathbf{H}_\mathrm{ext}$ in $\mathcal{C}$. Loops that wind around specific points in $\mathcal{C}$ induce colloidal motion. That is, once the loop returns to its initial position, the colloidal particle has moved to a different unit cell of the pattern. The transport is topologically protected since the precise form of the loop is irrelevant. Only the winding numbers of the loop around the specific points in $\mathcal{C}$ (which are the topological invariants) determine the transport.

### Experiments

The magnetic films with the desired patterns imprinted are thin Co/Au multilayers with perpendicular magnetic anisotropy[68] lithographically patterned via a home-built[69] keV-He-ion bombardment[70]. Further details about the fabrication process can be found in refs. 37,71–73.

The patterns have lattice vectors of magnitude 14 μm if not stated otherwise.

To reduce the influence of lateral magnetic field fluctuations due to the fabrication procedure (which increases near the substrate) we coat the magnetic pattern with a photo-resist film (thickness 1.6 μm). The coating layer serves other two purposes: it protects the patterns and it acts as a spacer between the colloidal particles and the pattern (see Fig. 1), in order to secure the condition $|\mathbf{H}_\mathrm{ext}| \gg |\mathbf{H}_\mathrm{p}|$. We then place paramagnetic colloids of diameter 2.8 μm immersed in deionized water on top of the pattern. The microparticles sediment and are suspended roughly the Debye length above the negatively charged coating layer on the pattern. The motion above the pattern is effectively two-dimensional.

The uniform external magnetic field is generated with three coils arranged around the pattern and controlled with a computer. The magnitude of the external field is approximately $4 \times 10^3$ A/m. Standard reflection microscopy techniques are used to visualize both the colloids and the pattern.

### Square and hexagonal periodic patterns

Consider magnetic periodic $N$-fold symmetric patterns with either $N = 2$ (square patterns) or $N = 3$ (hexagonal patterns). Examples of both types are shown in Supplementary Fig. 1. In the limit of an infinitely thin pattern located at $z = 0$, the magnetization is

$$\mathbf{M}(\mathbf{r}) = M(\mathbf{r}_\perp)\delta(z)\hat{\mathbf{e}}_z, \tag{3}$$

with $\delta(\cdot)$ the Dirac distribution, $\hat{\mathbf{e}}_z$ the unit vector normal to the pattern, $\mathbf{r}_\perp = (x, y)$, and

$$M(\mathbf{r}_\perp) = m_\mathrm{p}\,\mathrm{sign}\left(\sum_{i=1}^{N} \cos(\mathbf{q}_i \cdot (\mathbf{r}_\perp - \mathbf{b}) - \phi) + m_0(\phi)\right), \tag{4}$$

where $m_\mathrm{p}$ is the saturation magnetization of the domains. The wave vectors $\mathbf{q}_i$ in the square patterns are

$$\mathbf{q}_i = q_0 \begin{pmatrix} -\sin(\pi i/2 - \psi) \\ \cos(\pi i/2 - \psi) \end{pmatrix}, \quad i = 1,2 \tag{5}$$

with magnitude $q_0 = 2\pi/a$ and $a$ being the magnitude of a lattice vector, which in square patterns can be defined with the wave vectors being the reciprocal lattice vectors. That is, $\mathbf{a}_i \cdot \mathbf{q}_j = 2\pi\delta_{ij}$ (see Supplementary Fig. 1b). The global phase $\psi$ sets the orientation of the lattice vectors with respect to a fixed laboratory frame.

In the hexagonal patterns, the wave vectors are

$$\mathbf{q}_i = q_0 \begin{pmatrix} -\sin(2\pi i/3 - \psi) \\ \cos(2\pi i/3 - \psi) \end{pmatrix}, \quad i = 1,2,3 \tag{6}$$

with magnitude $q_0 = 4\pi/(a\sqrt{3})$. Here, the three wave vectors can be related to three (linearly dependent) lattice vectors via $\mathbf{q}_i \cdot \mathbf{a}_j = 2\pi\delta_{ij}$ for $i = 1, 2$ and $\mathbf{a}_3 \cdot \mathbf{q}_3 = 0$ (see Supplementary Fig. 1b).

In both square and hexagonal patterns, the wave vectors point into the $N$ different symmetry directions. The translational vector $\mathbf{b}$ in Eq. (4) plays a relevant role only in inhomogeneous patterns. In periodic patterns, we usually set $\mathbf{b} = \mathbf{0}$.

In square patterns, the symmetry phase $\phi$ in the magnetization (see Eq. (4)), simply causes a trivial shift of all Wigner−Seitz cells with respect to the origin of the pattern. Hence, for simplicity, we set it to zero. In hexagonal patterns however, the symmetry phase $\phi$ has a non-trivial effect since it determines the point symmetry of the pattern (see Supplementary Fig. 1c), and therefore the modulation loops required to transport the colloidal particles[37]. The Wigner−Seitz cell of a hexagonal pattern contains in general three symmetry points with C$_3$ symmetry (rotation through an angle $2\pi/3$ about the symmetry axis). For special values of the symmetry phase, one of the three-fold

symmetric points acquires a higher symmetry; either six-fold hexagonal $C_6$ symmetry (for $\phi = 0$ and $\phi = \pm\pi/3$) or $S_6$ symmetry, i.e. a $C_6$ followed by a perpendicular reflection (for $\phi = \pm\pi/6$).

Finally, the parameter $m_0$ in Eq. (4), which is actually a function of the symmetry phase $\phi$, alters the area ratio between up-magnetized and down-magnetized domains. Following Loehr et al.[37], we use here $m_0(\phi) = \frac{1}{2}\cos(3\phi)\delta_{N,3}$ (therefore in square patterns $m_0 = 0$) to ensure that the average magnetization in hexagonal patterns is very small, i.e.

$$\int M(\mathbf{r}_\perp)\mathrm{d}\mathbf{r}_\perp \approx 0. \tag{7}$$

### Magnetic field of the pattern

To numerically compute the magnetic field of the pattern, $\mathbf{H}_\mathrm{p}(\mathbf{r})$, at the desired position in action space we first discretize the pattern in a square grid with resolution $0.03a$ and compute the magnetization at the grid points via Eq. (4). Next, we compute the magnetic field at the grid points by convolution of the magnetization with the Green's-function of the system:

$$\mathbf{H}_\mathrm{p}(\mathbf{r}) = \mathbf{H}_\mathrm{p}(\mathbf{r}_\perp, z) = \frac{1}{4\pi}\int \mathrm{d}\mathbf{r}'_\perp \frac{\mathbf{r}_\perp - \mathbf{r}'_\perp + z\hat{\mathbf{e}}_z}{|\mathbf{r}_\perp - \mathbf{r}'_\perp + z\hat{\mathbf{e}}_z|^3} M(\mathbf{r}'_\perp). \tag{8}$$

Here $\mathbf{r}_\perp = (x, y)$ is the position coordinate in a plane parallel to the pattern. We calculate the magnetic field at an elevation above the pattern $z = 0.5a$, which is comparable to the experimental value. As usual, we perform the convolution in Fourier space.

To calculate the magnetic field at a generic, off-grid, position we simply interpolate the magnetic field using bicubic splines.

### Pattern with a topological defect

For the pattern with a topological defect shown in Fig. 2, the symmetry phase varies with the position $\mathbf{r}_\perp$ as

$$\phi(\mathbf{r}_\perp) = \frac{1}{3}\left(\frac{\pi}{2} - \arctan\left(\mathbf{q}_3 \cdot \mathbf{r}_\perp, \hat{\mathbf{e}}_z \cdot (\mathbf{r}_\perp \times \mathbf{q}_3)\right)\right), \tag{9}$$

and the global orientational phase is set to $\psi = 0$ in Eq. (6). For our choice of wave vectors (see Eq. (6) and Supplementary Fig. 1b), the symmetry phase modulation is simply $\phi(\mathbf{r}_\perp) = (\pi/2 - \arctan(x,y))/3$. Here $\arctan(y,x)$ returns the four-quadrant inverse tangent of $y/x$. The symmetry phase varies therefore between $\phi = -\pi/3$ and $\pi/3$ as we wind once around the origin. The topological charge of the defect located at the center of the pattern ($\mathbf{r}_\perp = 0$) is $q = \Delta\phi/(2\pi/p) = 1$. Here $\Delta\phi = 2\pi/3$ is the angle that the director rotates if we wind once counter-clockwise around the defect, and $p = 3$ is the $p$-atic symmetry of the director field[74]. (The symmetry phase can be described with a 3-atic director field for which the local orientations are defined modulo $\pi/3$.) Varying the symmetry phase between $-\pi/3$ and $\pi/3$ also introduces a shift of the unit cell, cf. the unit cells for $\phi = \pi/3$ and $-\pi/3$ in Supplementary Fig. 1c. To rectify this shift and avoid therefore discontinuities in the magnetization of the pattern, we need to use a local shift vector in Eq. (4) given by

$$\mathbf{b}(\mathbf{r}_\perp) = -(\mathbf{a}_1 + \mathbf{a}_2)\frac{\phi(\mathbf{r}_\perp)}{2\pi}. \tag{10}$$

The shift vector can be understood as a Burgers vector since it corrects for the spatial distortion of the pattern around the defect.

### Symmetry phase modulated patterns

To encode in the pattern the desired particle trajectories, we use the drawing software Krita[75]. We prescribe the stable trajectory on a square image with a side-length of 1000 pixels. In Krita, we draw the desired trajectory with a brush (thickness 1 pixel) that encodes the drawing direction in the hue of the colored pixels. The drawing direction directly translates into the transport direction that the particles will follow above the pattern. This procedure results in an image that is essentially empty except for the trajectory lines. We then map from hue to the symmetry phase $\phi$. An example of the pattern at this stage is shown in Supplementary Fig. 3a. The mapping from hue to $\phi$ is simply a linear transformation.

Next, we give a value to the symmetry phase everywhere in the pattern. To calculate the phase at a generic position $\mathbf{r}_\perp = (x, y)$ we average over all the prescribed phases along the trajectories. Each phase along the trajectory is weighted with a weight function proportional to $1/r_\mathrm{d}^2$, with $r_\mathrm{d}$ the distance between $\mathbf{r}_\perp$ and a point on the trajectory. Special care needs to be taken due to the periodicity of the symmetry phase[76]. We first transform the phases along the trajectories into unit vectors, next we average the vectors, and then transform back the averaged vector into a value of the symmetry phase. An illustration of the pattern after this stage is shown in Supplementary Fig. 3b. Finally, we use the value of the symmetry phase in the whole pattern to calculate the magnetization via Eq. (4) (see Supplementary Fig. 3c).

## Data availability

The code to simulate the system and to generate the modulation loops is available at Zenodo[67]. All other data supporting the findings are available from the corresponding author upon request.

## Code availability

A code to perform the adaptive Brownian Dynamics simulations of the colloidal particles as well as to generate the modulation loops is available at Zenodo[67].

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

## Acknowledgements

We thank Adrian Ernst for helping us to transfer the loops from the simulations to the experiment. We acknowledge funding by the Deutsche Forschungsgemeinschaft (DFG, German Research Foundation) under project numbers 440764520 (T.M.F. and D.d.l.H.) and 531559581 (D.d.l.H. and T.M.F.).

## Author contributions

N.C.X.S. designed the modulation loops and performed the computer simulations. F.F. performed the experiments. N.C.X.S., F.F., T.M.F., and D.d.l.H. conceptualized the research. P.K., F.S., and M.U. produced the magnetic film. S.A. and Ar.E. performed the fabrication of the micromagnetic domain patterns within the magnetic thin film. N.C.X.S., T.M.F., and D.d.l.H. designed the patterns and wrote the manuscript. All authors contributed to the different revision stages of the manuscript.

## Funding

## Competing interests

The authors declare no competing interests.
