## [Peer Review File · Nature Communications]

Simultaneous and independent topological control of identical microparticles in non-periodic energy landscapesREVIEWER COMMENTS

Reviewer #2 (Remarks to the Author):

The article by Stuhlmüller et al. is a combined experimental and simulation work that demonstrates the controlled transport of paramagnetic colloidal particles across periodic and more complex patterns by using loops of time-dependent uniform magnetic fields. The authors demonstrate a very powerful technique that allows to direct the motion of microscale colloids across a variety of complex patterns by using either simple or more intricate magnetic modulations. The authors show that a properly designed pattern with a central topological defect can act as attractant or repellent for microparticles in a rather robust way. Moreover, they show, and this is rather totally astonishing in the opinion of the Referee, the possibility to control simultaneously and independently several colloidal particles across these spatially complex patterns. The strategy adopted is to concatenate patches of a periodic square pattern and to program the loops in the control space such that the letters of the alphabet (see Fig.6b and Video in supporting info) can be simultaneously written. This strategy is somehow opposite than the previous one, where now complex is the pattern used (arrangement of patches) rather than the modulation loops in the control space. Finally, also the complex structure and complex modulation field allow to reach further degree of control on the motion of the magnetic beads.

All these results are beyond the status of the art in magnetic transport of colloidal matter, and they will have a strong impact in the field. For all these reasons, I recommend with emphasis this manuscript for Nature Communications.

I have a few comments related more on the way the manuscript is structured for the journal.

First, I find that the abstract and part of the description of previous work by the group in the introduction is too long and may deviate the attention of the reader from the novel results of this paper. Thus, I suggest shortening the abstract (first half of it is not needed) and part of the description of previous work.

The sections "System setup and computer simulation" and "Experiments" can also go in the method section (end of the manuscript) so they will not distract the reader from the main finding of the paper.

Also, I would comment more in the conclusion on the potential implications of this work to open new routes to program the motion of microscopic colloids. With this respect, the authors vaguely refer to the fact this transport technique leads to new "avenue in dynamic self-assembly" but it is related more on transport rather than assembly matter.

Reviewer #3 (Remarks to the Author):

The authors used a set of designing rules to create non-periodic magnetic patterns with defects which can transport paramagnetic colloidal particles to a desired position or a desired trajectory loop by repeating simple modulations. This method can simultaneously and independently drive particles at different initial positions to the same trajectory loop, as demonstrated by their simulations and experiments. It opens a new route to the dynamical control of colloidal systems which is likely to have various applications in the future. I recommend the publication for its outstanding novelty and importance. The authors should

address the comments below.

The authors have published the related works in refs.35, 36, 37, 48 which are actually different from the present work. The differences should be better explained in this manuscript.

Literature about applying the same designing rules to non-colloidal systems should be discussed and cited.

A concise qualitative explanation on the working principle without the terminologies of topological control would be helpful for readers in soft matter community.

In Fig.4, the orange particles 1-4 and blue particles 1-4 are different, right? Then they should be labelled by different numbers, or it seems that blue particle 1 is attracted to the defect and then repelled away as orange particle 1.

The Conclusion section is mainly about future outlooks; it should better summarize the results and conclusions, and compare with previous works.

To my knowledge, the author of ref.33 learned the idea and did the experiment in D.G. Grier's lab (e.g. coauthored PRB 66, 24504 (2002) etc.), but published their Nature paper without Grier! Thus, it is fair to cite some similar papers from Grier's group, e.g. PRL 89, 128301 (2003), PRE 70, 010901 (2004).

Minor points:

'spacer' usually refers to the object sandwiched between the two walls in the confinement of colloidal monolayer. Here spacer is actually a coating layer on the substrate, and thus it is better to be rephrased. The spheres in Fig.1a looks like inside instead of on top of the 'spacer'.

Page 8: "we draw the trajectories by hand..." Why not track particles' positions by image analysis, which is a standard analysis in colloid studies for 3 decades?

Reviewer #4 (Remarks to the Author):

Report for "Simultaneous and independent topological control of identical microparticles in non-periodic energy landscapes" by Stuhlmueller et al.

The authors report a colloidal system by whose motion can be precisely controlled by an underlying magnetic field which is set by a lattice. The key to the exquisite control that the authors demonstrate is their use of time-periodic loops of the magnetic field. The authors demonstrate this control with a number of examples, notably writing the letters of the alphabet. There is little doubt that this work marks a significant advance of a level certainly appropriate for Nature Communications. However, in its present form, in terms of clarity the manuscript falls far short of the level required. I am really struggling to gain anything more than the very sketchiest idea of how what the authors have done works. Which is a shame, as the results are really quite impressive.

My suggestion to the authors is to re-write the manuscript entirely, in a manner in keeping with the journal. I would suggest reference to papers in the field of driven colloids in Nature Communications as a guide to the level of clarity relevant here, in which this manuscript sadly falls very far short.

More specifically, I think the introduction is too long and overall there is too much material in the manuscript. The authors should describe, succinctly yet clearly, what they have done and how it stands out from what has been done previously.

One thing which should in particular be clarified is the connection with Ref 36, by the same authors in this same journal. How does this manuscript relate to the previous one? I see that there is some different symmetry in the lattice, but what are the consequences?

In addition to the introduction, the whole manuscript seems too long, by a factor of two, say (in addition to being rather impenetrable). I think that some material should be moved to the SI. The authors should carefully consider exactly what they want to say, precisely and concisely and in a manner which is accessible to the general audience of Nature Communications.

There also seems to be very little description of the computer simulations, other than they are Brownian dynamics. How are the magnetic fields implemented? Curiously, in their conclusions, the authors argue that Brownian motion is not important. This seems inconsistent with their use of BD simulations. Furthermore, at some point the Brownian nature of the particles must be, at some point become important. There must be a particle size where the amount of magnetic material that can be incorporated is not enough for the forces discussed here to overcome the Brownian motion. That size could I suppose be rather small, but the authors should at least mention it, rather than dismissing Brownian motion as they currently do.

A small point:

On p3, we read that

“The microparticles sediment and hover roughly the Debye length above the negatively charged spacer layer on the pattern.”

Colloids don't “hover”. Helicopters hover. Please replace “hover” by “are suspended”.

We would like to thank the Reviewers for their work assessing the manuscript, for providing constructive criticism, and for the positive evaluation of our work. We give below a detailed response to each point. In addition, we include in the resubmission a version of the manuscript with the changes highlighted in blue.

Reviewer #2 (Remarks to the Author):

The article by Stuhlmüller et al. is a combined experimental and simulation work that demonstrates the controlled transport of paramagnetic colloidal particles across periodic and more complex patterns by using loops of time-dependent uniform magnetic fields. The authors demonstrate a very powerful technique that allows to direct the motion of microscale colloids across a variety of complex patterns by using either simple or more intricate magnetic modulations. The authors show that a properly designed pattern with a central topological defect can act as attractant or repellent for microparticles in a rather robust way. Moreover, they show, and this is rather totally astonishing in the opinion of the Referee, the possibility to control simultaneously and independently several colloidal particles across these spatially complex patterns. The strategy adopted is to concatenate patches of a periodic square pattern and to program the loops in the control space such that the letters of the alphabet (see Fig.6b and Video in supporting info) can be simultaneously written. This strategy is somehow opposite than the previous one, where now complex is the pattern used (arrangement of patches) rather than the modulation loops in the control space. Finally, also the complex structure and complex modulation field allow to reach further degree of control on the motion of the magnetic beads.

All these results are beyond the status of the art in magnetic transport of colloidal matter, and they will have a strong impact in the field. For all these reasons, I recommend with emphasis this manuscript for Nature Communications.

We thank the Reviewer for their careful work in reviewing our paper and for the efforts producing a report. We are very pleased by the positive judgment and we acknowledge the points that the Referee raises. We have made corresponding changes to the paper, as laid out in detail in the responses to each point below.

I have a few comments related more on the way the manuscript is structured for the journal.

First, I find that the abstract and part of the description of previous work by the group in the introduction is too long and may deviate the attention of the reader from the novel results of this paper. Thus, I suggest shortening the abstract (first half of it is not needed) and part of the description of previous work.

We agree with the Reviewer and have shortened the abstract from 254 words to 161.

We have also replaced a significant part of the content in the introduction with a brief description of previous work and moved the detailed explanation of the transport in periodic patterns to a new Supplementary information.

The sections "System setup and computer simulation" and "Experiments" can also go in the method section (end of the manuscript) so they will not distract the reader from the main finding of the paper.

We agree and have moved both sections accordingly.

Also, I would comment more in the conclusion on the potential implications of this work to open new routes to program the motion of microscopic colloids. With this respect, the authors vaguely refer to the fact this transport technique lead to new "avenue in dynamic self-assembly" but it is related more on transport rather than assemble matter.

In the revised version, we mention in the abstract and also in the Introduction that our work also opens new avenues in transport control.

We believe that our work is also relevant for dynamically controlling the self-assembly of colloidal particles. In the conclusions of the revised version, we give a brief example of current work that goes in that direction [54]. Using inhomogeneous patterns and complex modulation loops, we move individual isotropic colloidal particles toward a polymerization site. There, the particles assemble one-by-one into rods, waiting until the rod has the desired length. Once the final length is achieved, the rod walks away from the polymerization site following the desired trajectory. The new text in the Discussion reads:

"Therefore, beyond offering the possibility to control the transport of identical microparticles simultaneously, our work also opens a new route towards dynamical self-assembly in colloidal science. As an example, we have created a colloidal rod factory [54] in which identical isotropic particles are transported towards a reaction site in which they self-assemble. Only when they reach the desired aspect ratio, the rods leave the polymerization site following the desired trajectory. The use of patchy colloids [55-58] with e.g. hybridization of complementary DNA strands [59-61] and other shape-anisotropic particles [62, 63] would offer more versatility to create complex functional structures."

Added references:

[54] J. Elschner, F. Farrokhzad, P. Kuświk, M. Urbaniak, F. Stobiecky, S. Akhundzada, A. Ehresmann, D. de las Heras, and T. M. Fischer, Topologically controlled synthesis of active colloidal bipeds, In preparation (2023).

[62] C. J. Hernandez and T. G. Mason, Colloidal alphabet soup: monodisperse dispersions of shape-designed LithoParticles, J. Phys. Chem. C 111, 4477 (2007)

[63] S. Sacanna, D. J. Pine, and G.-R. Yi, Engineering shape: the novel geometries of colloidal self-assembly, Soft Matter 9, 8096 (2013)

Reviewer #3 (Remarks to the Author):

The authors used a set of designing rules to create non-periodic magnetic patterns with defects which can transport paramagnetic colloidal particles to a desired position or a desired trajectory loop by repeating simple modulations. This method can simultaneously and independently drive particles at different initial positions to the same trajectory loop, as demonstrated by their simulations and experiments. It opens a new route to the dynamical control of colloidal systems which is likely to have various applications in the future. I recommend the publication for its outstanding novelty and importance. The authors should address the comments below.

We would like to thank the Reviewer for a thorough reading of our paper, for the positive recommendation, and also for making valuable suggestions.

The authors have published the related works in refs. 35, 36, 37, 48 which are actually different from the present work. The differences should be better explained in this manuscript.

We have moved the extended description of the transport in periodic patterns (which is part of our previous work) to a new Supplementary Information.

To clarify the differences, we have also rewritten a substantial part of the Introduction and created a new figure 1. The new figure illustrates the transport in periodic patterns (old work) and in inhomogeneous patterns (this work).

We also explain briefly in the introduction the limitations of the transport in periodic patterns and how this work overcomes them.

Literature about applying the same designing rules to non-colloidal systems should be discussed and cited.

We are not aware of other non-colloidal systems in which topological protection has been used to transport particles independently and simultaneously. However, we believe that our ideas can be at least partially transferable to other systems in which the transport is based on topological invariants. We refer now in the discussion to several other works in which topology is at work.

"Our ideas might be transferable to other systems in which the transport is also based on topological protection. These include, solitons [45], nano-machines [46, 47], sound waves [48, 49], photons [50, 51], and quantum mechanical excitations [52]."

We also refer in the Discussion to a recent work that follows a different approach to achieve a related goal:

"The complexity of the transport is encoded [in our work] in the magnetic potential which varies in space and in time via the magnetic patterns and the modulation loops, respectively. An alternative approach that encodes the transport in the particle shape has appeared recently [53]. There, Sobolev et al. find the shape of the rigid-body that traces the desired trajectory when rolling down a slope."

New references:

[45] B. G. Ge Chen, N. Upadhyaya, and V. Vitelli, Nonlinear conduction via solitons in a topological mechanical insulator, Proc. Natl. Acad. Sci. 111, 13004 (2014)

[46] M. Porto, M. Urbakh, and J. Klafter, Atomic scale engines: Cars and wheels, Phys. Rev. Lett. 84, 6058 (2000)

[47] V. L. Popov, Nanomachines: Methods to induce a directed motion at nanoscale Phys. Rev. E 68, 026608 (2003)

[48] L. M. Nash, D. Kleckner, A. Read, V. Vitelli, A. M. Turner, and W. T. M. Irvine, Topological mechanics of gyroscopic metamaterials,

Proc. Natl. Acad. Sci. 112, 14495 (2015).

[49] Z. Yang, F. Gao, X. Shi, X. Lin, Z. Gao, Y. Chong, and B. Zhang, Topological acoustics, Phys. Rev. Lett. 114, 114301 (2015).

[50] S. Mittal, J. Fan, S. Faez, A. Migdall, J. Taylor, and M. Hafezi, Topologically robust transport of photons in a synthetic gauge field, Phys. Rev. Lett. 113, 087403 (2014).

[51] J. Ningyuan, C. Owens, A. Sommer, D. Schuster, and J. Simon, Time- and site-resolved dynamics in a topological circuit, Phys. Rev. X 5, 021031 (2015).

[52] M. Z. Hasan and C. L. Kane, Colloquium: Topological insulators, Rev. Mod. Phys. 82, 3045 (2010).

[53] Y. I. Sobolev, R. Dong, T. Tlustý, J.-P. Eckmann, S. Granick, and B. A. Grzybowski, Solid-body trajectoids shaped to roll along desired pathways, Nature 620, 310 (2023).

A concise qualitative explanation on the working principle without the terminologies of topological control would be helpful for readers in soft matter community.

In the rewritten Introduction and newly added figure 1, we explain the working principle with hopefully simple terms and reducing as much as possible the terminology of topological control.

Also, in the beginning of the Results section we give further hints and point to the extended description of the transport in periodic patterns which is now given as Supplementary Information.

In Fig.4, the orange particles 1-4 and blue particles 1-4 are different, right? Then they should be labelled by different numbers, or it seems that blue particle 1 is attracted to the defect and then repelled away as orange particle 1.

We agree with the Reviewer and have changed the figure accordingly. The blue and orange trajectories were measured in two experimental realizations and have been superimposed in the figure. We mention this in the caption of Fig. 3:

"Blue and orange trajectories correspond to different experiments and have been superimposed in the figure."

The Conclusion section is mainly about future outlooks; it should better summarize the results and conclusions, and compare with previous works.

We have substantially revised the Discussion (see text highlighted in blue) and included a summary of the results. A comparison with our previous works is now presented in the Introduction.

To my knowledge, the author of ref.33 learned the idea and did the experiment in D.G. Grier's lab (e.g. coauthored PRB 66, 24504 (2002) etc.), but published their Nature paper without Grier! Thus, it is fair to cite some similar papers from Grier's group, e.g. PRL 89, 128301 (2003), PRE 70, 010901 (2004).

We were not aware of these circumstances. We now cite both works suggested by the Reviewer (see new Refs. 33 and 35).

[33] P. T. Korda, M. B. Taylor, and D. G. Grier, Kinetically locked-in colloidal

transport in an array of optical tweezers, Phys. Rev. Lett. 89, 128301 (2002).

[35] K. Ladavac, K. Kasza, and D. G. Grier, Sorting mesoscopic objects with periodic potential landscapes: Optical fractionation, Phys. Rev. E 70, 010901 (2004).

Minor points:

'spacer' usually refers to the object sandwiched between the two walls in the confinement of colloidal monolayer. Here spacer is actually a coating layer on the substrate, and thus it is better to be rephrased.

The Reviewer is right that the spacer is actually a coating layer on the substrate. However, it serves several purposes, the most important one being acting as a spacer between the patterns and the colloidal particles. A particle-pattern vertical separation of the order of one unit cell is required to secure the condition $|H_{ext}| \gg |H_p(r_A)|$ which simplifies the form of the magnetic potential. We have clarified the role of the polymer coating in the caption of Fig. 1

"A polymer coating protects the patterns and acts as a spacer..."

and also in the Methods section:

"The coating layer serves other two purposes: it protects the patterns and it acts as a spacer between the colloidal particles and the pattern, see Fig. 1, in order to secure the condition $|H_{ext}| \gg |H_p(r_A)|$."

The spheres in Fig.1a looks like inside instead of on top of the 'spacer'.

We have replaced Fig. 1 with a new figure. We hope the spheres now give the impression of being on top of the spacer. The old Fig. 1 is now present in the Supplementary Information since it is still relevant to understand the transport in periodic patterns. We have also changed the elevation of the spheres there.

Page 8: "we draw the trajectories by hand..." Why not track particles' positions by image analysis, which is a standard analysis in colloid studies for 3 decades?

We have not expressed the correct idea. We do track the particles automatically. We meant that to create patterns like that in Fig. 4b the starting point is to draw by hand the trajectories that we want the particles to follow. We have clarified this point in page 5:

"In essence, we first draw the trajectories that the particles should follow by hand. Then, at each position along the trajectory we encode the transport direction using the value of the symmetry phase."

Also, the complete procedure to generate the patterns is described in Methods section and illustrated in Supplementary Fig. 3.

Reviewer #4 (Remarks to the Author):

Report for "Simultaneous and independent topological control of identical microparticles in non-periodic energy landscapes" by Stuhlmüller et al.

The authors report a colloidal system by whose motion can be precisely controlled by an underlying magnetic field which is set by a lattice. The key to the exquisite control that the authors demonstrate is their use of time-periodic loops of the magnetic field. The authors demonstrate this control with a number of examples, notably writing the letters of the alphabet. There is little doubt that this work marks a significant advance of a level certainly appropriate for Nature Communications. However, in its present form, in terms of clarity the manuscript falls far short of the level required. I am really struggling to gain anything more than the very sketchiest idea of how what the authors have done works. Which is a shame, as the results are really quite impressive.

My suggestion to the authors is to re-write the manuscript entirely, in a manner in keeping with the journal. I would suggest reference to papers in the field of driven colloids in Nature Communications as a guide to the level of clarity relevant here, in which this manuscript sadly falls very far short.

More specifically, I think the introduction is too long and overall there is too much material in the manuscript. The authors should describe, succinctly yet clearly, what they have done and how it stands out from what has been done previously.

We thank the Reviewer for the positive judgment of the results and for providing constructive criticism. As stated below we have made changes to improve the clarity of the manuscript (see our response to the point "In addition to the introduction..." below).

One thing which should in particular be clarified is the connection with Ref 36, by the same authors in this same journal. How does this manuscript relate to the previous one? I see that there is some different symmetry in the lattice, but what are the consequences?

The old reference [36] (Ref. [39] in the new version of the manuscript) describes the transport in a periodic hexagonal pattern (C6 symmetry) of diamagnets and paramagnets. Both types of particles belong to different topological classes and therefore can be transported in different directions simultaneously. There are several differences between both works. Here, we use inhomogeneous (non-periodic) patterns which, contrary to [36], allows us to:

- Transport identical particles in different directions simultaneously
- Make it possible that the particles follow different trajectories depending on their position above the patterns (in [39] identical particles move in the same direction independently of their position above the pattern)
- Transport a particle initially located in an unknown position to the desired position in the pattern.

We now mention the differences between the transport in periodic and non-periodic patterns in the introduction and also illustrate them in the new Fig. 1.

"The specific orientations of the external field that are relevant to control the motion depend on both the symmetry of the pattern [37] (e.g. square vs hexagonal) and the particle properties. Hence, particles with different properties, e.g. paramagnetic and diamagnetic particles above hexagonal magnetic patterns [39] ... can be transported in different directions independently and simultaneously ..."

"However, the use of periodic patterns imposes several limitations to the transport. All particles ..."

"These limitations are overcome here using inhomogeneous (non-periodic) patterns. We make either the symmetry, Fig. 1(b), or the global orientation, Fig. 1(c), of the magnetic pattern dependent on the absolute position above the pattern. As a result, the specific orientations of the external field that control the motion depend also on the space coordinate. The direction of the transport can then be locally controlled by the modulation loop of the external field and also via the local symmetry of the inhomogeneous magnetic pattern.

In addition to the introduction, the whole manuscript seems too long, by a factor of two, say (in addition to being rather impenetrable). I think that some material should be moved to the SI. The authors should carefully consider exactly what they want to say, precisely and concisely and in a manner which is accessible to the general audience of Nature Communications.

We agree with the Reviewer that the content of the paper, specially the description of the transport in periodic patterns (previous work) was too long. We have reduced the abstract and moved substantial parts of the main text (including figures) to either the Supplementary Information or the Methods section. The current word count of the main text (not including Abstract, Methods, References and Figure legends) is less than 4000 words, which is well below the Nature Communications limit of 5000 words. While we have not rewritten the manuscript entirely, we have substantially revised the Introduction and the Discussion. We believe that the reduction of the text together with the addition of the schematic figure 1, and the changes performed to the Introduction, the Results, and the Discussion have made the manuscript more accessible to the broad audience of the journal.

There also seems to be very little description of the computer simulations, other than they are Brownian dynamics. How are the magnetic fields implemented? Curiously, in their conclusions, the authors argue that Brownian motion is not important. This seems inconsistent with their use of BD simulations. Furthermore, at some point the Brownian nature of the particles must be, at some point become important. There must be a particle size where the amount of magnetic material that can be incorporated is not enough for the forces discussed here to overcome the Brownian motion. That size could I suppose be rather small, but the authors should at least mention it, rather than dismissing Brownian motion as they currently do.

We describe the calculation of the magnetic field of the pattern in Methods (see labels: "Square and hexagonal periodic patterns" and also "Magnetic field of the pattern"). Once the magnetic field of the pattern is known, the simulations are straight forward. The particles are subject to the magnetic potential given by Eq. (1). The equation of motion, Eq. (2), is then integrated in time. We believe the Methods section gives enough details to reproduce the simulations.

Nevertheless, to provide full details, we have included with this resubmission a complete code to simulate the particle transport and to generate the modulation loops. The code is accessible via Zenodo and it is referenced in both the Methods section and the added Code Availability statement.

Link to the code:

<https://zenodo.org/record/8413928>

Regarding the effect of Brownian motion, the equation of motion, Eq. (2), is adequate to model the system because the experiments are overdamped. That is, inertial effects do not play any role. The overdamped character of the motion is

the reason behind the use of (overdamped) Brownian dynamics simulations. Note that this is not incompatible with the almost absence of Brownian motion in the experiments (as one can observe in the videos). We have previously shown that the transport is robust in presence of Brownian motion using computer simulations [45]. Here, we have also tested the robustness of the transport against the occurrence of Brownian motion (see Supplementary Figure 4).

We have clarified these points in Methods (label: System setup and computer simulations)

"In the experiments, the magnetic forces dominate over the random forces which therefore do not play any role. We use Brownian dynamics simulations due to the overdamped character of the motion in the viscous aqueous solvent. The code to simulate the colloidal motion and to generate the modulation loops is available via Zenodo [67]"

and in the Discussion:

"Since the transport is topologically protected, it is robust against perturbations such as the presence of Brownian motion [44]. If we make Brownian motion more prominent (e.g. by increasing the temperature or reducing the particle size) the particles start to deviate from the expected trajectories but overall the transport is robust. An example of Brownian dynamics simulations at different temperatures is shown in Supplementary Fig. 4. The topological protection will disappear due to Brownian motion at sufficiently high temperatures and for small enough particles. A possible solution would then be to increase the magnitude of either the pattern field or the external magnetic field such that the magnetic forces dominate again the transport.

A small point:

On p3, we read that

"The microparticles sediment and hover roughly the Debye length above the negatively charged spacer layer on the pattern."

Colloids don't "hover". Helicopters hover. Please replace "hover" by "are suspended"

We agree and have changed the text accordingly. The new sentence (now in Methods) reads:

"The microparticles sediment and are suspended roughly the Debye length above the negatively charged coating layer on the pattern"

We also mention that the colloids "are suspended" in the caption of the new figure 1.

Further changes

Added Reference

[43] R. S. Hendley, L. Zhang, and M. A. Bevan,

Multistate dynamic pathways for anisotropic colloidal assembly and reconfiguration, ACS Nano 10.1021/acsnano.3c07202 (2023)

Correction of minor typos and clarifications.

Added statements of Data Availability, Code Availability, and Competing Interest.

REVIEWERS' COMMENTS

Reviewer #2 (Remarks to the Author):

The authors of the manuscript have carefully take into account all my suggestions. The work has now improved and I'm happy to support publication in Nature Comm.

Reviewer #3 (Remarks to the Author):

My concerns have been well addressed. I recommend the publication.

Reviewer #4 (Remarks to the Author):

Second report for Report for “Simultaneous and independent topological control of identical microparticles in non-periodic energy landscapes” by Stuhlmüller et al.

Following the first round of reviewing, the authors have very substantially revised their manuscript. In particular, the text is very much clearer, and they have substantially shortened the manuscript. They have furthermore addressed a number of more specific points raised, such as the relationship of this manuscript with their previous work, and the discussion of Brownian motion. The latter does still need some small adjustment: the authors currently state that their experiments are in the zero temperature limit where Brownian motion is negligible. Very obviously, the experiments are not performed at zero temperature. If the authors can reword this sentence then that would be fine.

Smaller points and typos:

(1) I think it would read better to delete “then” in the abstract in the clause “...when then microparticles are subjected to simple time-periodic loops.”

(2) Results, line 4: boundary → boundaries.

(3) p3, first column line 2: patters → patterns.

(4) p3, “The symmetry phase is constant along radial rays.....”

I don't really understand what is meant by “rays” here. Do the authors mean “directions”?

Reviewer #2 (Remarks to the Author):

The authors of the manuscript have carefully take into account all my suggestions. The work has now improved and I'm happy to support publication in Nature Comm.

We thank the Reviewer for their positive recommendation.

Reviewer #3 (Remarks to the Author):

My concerns have been well addressed. I recommend the publication.

We thank the Reviewer for their positive recommendation.

Reviewer #4 (Remarks to the Author):

Second report for Report for "Simultaneous and independent topological control of identical microparticles in non-periodic energy landscapes" by Stuhlmüller et al.

Following the first round of reviewing, the authors have very substantially revised their manuscript. In particular, the text is very much clearer, and they have substantially shortened the manuscript. They have furthermore addressed a number of more specific points raised, such as the relationship of this manuscript with their previous work, and the discussion of Brownian motion. The latter does still need some small adjustment: the authors currently state that their experiments are in the zero temperature limit where Brownian motion is negligible. Very obviously, the experiments are not performed at zero temperature. If the authors can reword this sentence then that would be fine.

We thank the Reviewer for a careful reading of the manuscript, the constructive criticism and the positive recommendation. We have addressed the points raised by the reviewer:

We have reworded the sentence which now reads:

"In the experiments, the Brownian motion of the colloidal particles is negligible but it might play a role in other systems with smaller colloids and/or at higher temperatures."

Smaller points and typos:

(1) I think it would read better to delete "then" in the abstract in the clause "...when then microparticles are subjected to simple time-periodic loops."

We have corrected the typo. The new sentence reads:

"... when the microparticles are subjected ..."

(2) Results, line 4: boundary → boundaries.

(3) p3, first column line 2: patters → patterns.

Thank you, we have corrected both typos.

(4) p3, "The symmetry phase is constant along radial rays...."

I don't really understand what is meant by "rays" here. Do the authors mean "directions"?

Yes, we meant directions. We have updated the text accordingly.